Green synthesis of silver nanoparticles in aloe vera plant extract prepared by a hydrothermal method and their synergistic antibacterial activity

Tippayawat Patcharaporn patchatip@kku.ac.th 1 2
Phromviyo Nutthakritta 3
Boueroy Parichart 4
Chompoosor Apiwat 5 6
1 Division of Clinical Microbiology/Faculty of Associated Medical Sciences, Khon Kaen University , Khon Kaen , Thailand
2 The Center for Research & Development of Medical Diagnostic Laboratories/Faculty of Associated Medical Sciences, Khon Kaen University , Khon Kaen , Thailand
3 Materials Science and Nanotechnology Program/Faculty of Science, Khon Kaen University , Khon Kaen , Thailand
4 Department of Microbiology/Faculty of Medicine, Khon Kaen University , Khon Kaen , Thailand
5 Department of Chemistry/Faculty of Science, Ramkhamhaeng University , Bangkok , Thailand
6 Integrated Nanotechnology Research Center (INRC), Khon Kaen University , Khon Kaen , Thailand
Corbo Maria Rosaria
Electronic publication date: 2016 Oct 19
Publication date: 2016
Volume: 4
Electronic Location ID: e2589
Received 2016 Mar 29; Accepted 2016 Sep 20
Copyright: ©2016 Tippayawat et al.
Copyright year: 2016
Copyright holder: Tippayawat et al.
License: This is an open access article distributed under the terms of the Creative Commons Attribution License, which permits unrestricted use, distribution, reproduction and adaptation in any medium and for any purpose provided that it is properly attributed. For attribution, the original author(s), title, publication source (PeerJ) and either DOI or URL of the article must be cited.
License URL: https://creativecommons.org/licenses/by/4.0/

Keywords: Hydrothermal, Green synthesis, Aloe vera, Antibacterial activity, Silver nanoparticles

Funding: Thailand Research Fund and Khon Kaen University TRG5880024 Integrated Nanotechnology Research Center (INRC) This work was supported by the Thailand Research Fund and Khon Kaen University (TRG5880024) and Integrated Nanotechnology Research Center (INRC), Khon Kaen University. The funders had no role in study design, data collection and analysis, decision to publish, or preparation of the manuscript.

==============================
Background

There is worldwide interest in silver nanoparticles (AgNPs) synthesized by various chemical reactions for use in applications exploiting their antibacterial activity, even though these processes exhibit a broad range of toxicity in vertebrates and invertebrates alike. To avoid the chemical toxicity, biosynthesis (green synthesis) of metal nanoparticles is proposed as a cost-effective and environmental friendly alternative. Aloe vera leaf extract is a medicinal agent with multiple properties including an antibacterial effect. Moreover the constituents of aloe vera leaves include lignin, hemicellulose, and pectins which can be used in the reduction of silver ions to produce as AgNPs@aloe vera (AgNPs@AV) with antibacterial activity.

Methods

AgNPs were prepared by an eco-friendly hydrothermal method using an aloe vera plant extract solution as both a reducing and stabilizing agent. AgNPs@AV were characterized using XRD and SEM. Additionally, an agar well diffusion method was used to screen for antimicrobial activity. MIC and MBC were used to correlate the concentration of AgNPs@AV its bactericidal effect. SEM was used to investigate bacterial inactivation. Then the toxicity with human cells was investigated using an MTT assay.

Results

The synthesized AgNPs were crystalline with sizes of 70.70 ± 22-192.02 ± 53 nm as revealed using XRD and SEM. The sizes of AgNPs can be varied through alteration of times and temperatures used in their synthesis. These AgNPs were investigated for potential use as an antibacterial agent to inhibit pathogenic bacteria. Their antibacterial activity was tested on S. epidermidis and P. aeruginosa. The results showed that AgNPs had a high antibacterial which depended on their synthesis conditions, particularly when processed at 100 oC for 6 h and 200 oC for 12 h. The cytotoxicity of AgNPs was determined using human PBMCs revealing no obvious cytotoxicity. These results indicated that AgNPs@AV can be effectively utilized in pharmaceutical, biotechnological and biomedical applications.

Discussion

Aloe vera extract was processed using a green and facile method. This was a hydrothermal method to reduce silver nitrate to AgNPs@AV. Varying the hydrothermal temperature provided the fine spherical shaped nanoparticles. The size of the nanomaterial was affected by its thermal preparation. The particle size of AgNPs could be tuned by varying both time and temperature. A process using a pure AG phase could go to completion in 6 h at 200 oC, whereas reactions at lower temperatures required longer times. Moreover, the antibacterial effect of this hybrid nanomaterial was sufficient that it could be used to inhibit pathogenic bacteria since silver release was dependent upon its particle size. The high activity of the largest AgNPs might have resulted from a high concentration of aloe vera compounds incorporated into the AgNPs during hydrothermal synthesis.

Introduction

Silver nanoparticles (AgNPs) have been extensively studied for many decades due to their unique features and wide range of applications. Their uses include catalysis (Pradhan, Pal & Pal, 2002), biosensing (Anker et al., 2008), imaging (Lee & El-Sayed, 2006), and antibacterial activity (Morones et al., 2005; Rai, Yadav & Gade, 2009). Among these applications, antibacterial activities have gained much attention because they potentially offer a solution to the problem of antibiotic resistance (Cho et al., 2005). There are a variety of methods to synthesize AgNPs including physical and chemical methods (Chudasama et al., 2010). Chemical reduction of silver ions using sodium borohydride (Zhang et al., 2000), hydrazine (Taleb, Petit & Pileni, 1997), ascorbic acid (Lee et al., 2004), trisodium citrate (Sun, Mayers & Xia, 2003), and polyols (Sun & Xia, 2002) were reported and are considered well-established methods. Although chemical routes are effective, these methods may suffer from toxicity due to the chemicals used and the difficulty in removing them. Additionally, chemical reagents used in these methods are hazardous to the environment (Nabikhan et al., 2010). To avoid the toxicity of chemicals, green synthesis was developed (Sharma, Yngard & Lin, 2009). This method of biosynthesis of metal nanoparticles has been proposed as a costeffective and environmental friendly way of fabricating these materials.

Synthesis of AgNPs employing either microorganisms or plant extracts has emerged as an alternative approach. These biosynthetic methods have a numbers of benefits They are simple, cost-effective, give high yields, and are environmentally friendly (Zhang et al., 2013). Plant extracts have reportedly been used in the preparation of AgNPs (Sun et al., 2014). Aloe vera leaves have been used as medicinal plants since they possess anti-inflammatory activity, UV protection, antiarthritic properties, promote wound and burn-healing, and have antibacterial properties (Chandran et al., 2006; Feng et al., 2000; Reynolds & Dweck, 1999; Vazquez et al., 1996). There are a number of biologically active constituents in aloe vera leaves. These include lignin, hemicellulose, pectins which can be used in the reduction of silver ions (Emaga et al., 2008). It is believed that the large enzymes and proteins in aloe vera extract are weakly bound to silver ions and function as a complexing agent. Due to their low cost and environmentally friendly nature coupled with their reducing properties, we selected aloe vera as the reducing and stabilizing agent to prepare AgNPs and test their antibacterial activity.

In this study, we report a one-step hydrothermal method to prepare silver nanoparticles. Reduction of Ag+ ions to Ag0 nanoparticles was done in a medium of aloe vera extract in which no extra reducing agent was used. The resulting AgNPs can be obtained in large quantities. The sizes of AgNPs were found to be in a range of 70.70–192.02 nm and controllable by varying temperature and time conditions of the hydrothermal process. Further, the resulting AgNPs were found to be effective against gram-positive (Streptococcus epidermidis) and gram-negative (Pseudomonas aeruginosa). Therefore, this work has shown the use of naturally occurring compounds to be a reducing and stabilizing agent. This method is considered green synthesis. The resulting silver nanoparticles showed a synergism of aloe vera and silver nanoparticles on bactericidal effect. This hybrid nanomaterial provides an alternative material for using in antibacterials.

Materials and Methods

In this study, silver nitrate, AgNO3 (Sigma-Aldrich Chemicals, USA) and aloe vera plant extract were used as the starting materials. The aloe vera extract solution was prepared using 50 g of aloe vera leaves that had been rinsed with deionized water and finely cut into small pieces. The chopped aloe vera leaves were boiled in a 50 mL of deionized water for 20 minutes and allowed to cool. The cooled leaf broth was filtered and stored in a refrigerator at 4 °C. The resulting extract was used as an aloe vera extract solution.

Synthesis of AgNPs and characterization of AgNPs

In the preparation of AgNPs samples, AgNO3 (0.3 mol) was first dissolved in 20 ml of deionized water and mixed with 20 ml of aloe vera extract solution under vigorous stirring at room temperature for 30 min. The mixtures were added to sealed Teflon-lined vessels of 100 mL capacity (Parr, USA), which were heated and maintained at various time and temperature conditions, and then gradually cooled to room temperature. A gray precipitate was collected by filtration and washed with deionized water several times, and finally dried in air at 60 °C for 6 h. The crystal phase analysis of the AgNPs powders was conducted using X-ray diffraction (XRD) (PW3710, the Netherlands) with CuKα radiation (λ = 0.15406 nm). The particle sizes and morphology of the prepared AgNPs samples were characterized using scanning electron microscopy (SEM) (LEO SEM 1450VP, UK) and transmission electron microscopy (TEM) (FEI 5022/22 Tecnai G2 20 STwin, CR). The UV–visible absorbance of the AgNPs was measured using UV-1800 (Shimadzu, Japan).

Antibacterial tests and cytotoxicity test

Well diffusion method

The antibacterial activity of AgNPs prepared under different hydrothermal processing conditions were tested against gram-negative P. aeruginosa (Pseudomonas aeruginosa, ATCC27803) and gram-positive S. epidermidis (Staphylococcus epidermidis, ATCC35984) using an agar well diffusion method. The organisms were sub-cultured in nutrient broth at 37 °C and incubated overnight. After that, Nutrient Agar (Merck) was swabbed with the respective subcultures (1×108 CFU/ml). Specimens containing AgNPs were then arranged on the swabbed agar surface and incubated at 37 °C for 24 h. The results were read by measuring the diameter of the inhibition zone (mm). The experiments were done in triplicate.

Scanning electron microscopy (SEM)

Scanning electron microscopy of control cells and AgNPs treated cells (0.04 mg/mL) was performed to investigate the antibacterial activity. Each bacterial culture was prepared as described above and then pipetted into a 6-well plate with and without AgNPs prior to covering the wells with glass slides. After incubating at 37 °C overnight, the glass slides were removed and gently washed with phosphate buffer saline 3 times before dehydration in an alcohol series using concentrations of 25%, 50%, 75%, 90% and 100% ethanol in distilled water. The slides were left in each concentration for 20 min. They were then air dried and kept in a desiccator until analysis.

Minimum inhibitory concentration (MIC) and minimal bactericidal concentration (MBC)

A microdilution method was used to indicate the bactericidal effect of AgNPs. A suspension of 1×108 CFU/ml of bacteria in nutrient broth was prepared as described above. The antibacterial solutions were prepared using serial two-fold (1:2) dilutions of AgNPs in concentrations ranging from 0.04 to 0.00008 mg/mL and incubated at 37 °C for 24 h. In the range of sample turbidity, the MIC of the samples could not be determined to identify the lowest concentration of antibacterial agent that inhibits 99% of the growth of the bacteria. A microdilution measurement was done in triplicate to confirm the value of MIC for each tested bacteria. As such, the MBC was measured after MIC determination. In this assay, 10 μl from all concentrations of AgNPs were pipetted onto nutrient agar plates and incubated at 37 °C for 24 h. The MBC endpoint was interpreted at the lowest concentration of antibacterial agent killing 100% of the initial bacterial population.

Cytotoxicity test

The AgNP samples produced at 100 °C for 6 h and 200 °C for 12 h were tested for their cytotoxicity using the MTT (3-4,5-dimethylthiazol-2-yl)-2,5-diphenyltetrazolium bromide) assay. Human peripheral blood mononuclear cells (PBMCs) from the leftover buffy coat were suspended into complete 1640 RPMI (supplemented with 10% fetal bovine serum, 2 mM L-glutamine, 100 unit/ml penicillin and 100 µg/ml streptomycin) in a 96-well plate at a density of 105 cells/well. This was done prior to exposure to AgNPs dissolved in RPMI to make a stock concentration at 0.04 mg/mL. The stock solution was used to generate serial two-fold dilution at 4 concentrations, i.e., 0.02, 0.01, 0.005, and 0.0025 mg/mL. Then, the cells were incubated at 37 °C in a fully humidified, 5% CO2 air atmosphere for 48 h. The test samples were removed from the cell cultures and the cells were reincubated for a further 24 h in fresh medium. They were then tested using the MTT assay. Briefly, 50 μl of MTT in phosphate buffered saline at 5 mg/ml was added into a medium in each well and the cells were incubated for 4 h. The medium and MTT were then gently aspirated from the wells and solubilized in formazan with 200 μl of DMSO and 25μl of Sorense’s Glycine buffer, pH 10.5. The optical density was read with a microplate readr at a wavelength of 560 nm. The average of 3 wells was used to determine the mean of each point. Then % survival of the cells was calculated. For each test sample, the data was used to determine the concentration of sample required to kill 50% (IC50) of the cells compared to that of the controls. A dose–response curve was derived from 5 concentrations in the test range using 3 wells per concentration.

Results

Characterization of silver nanoparticles

The morphology of AgNPs prepared at different reaction temperatures and times was examined using SEM. The result showed SEM images of AgNPs obtained by the reduction of AgNO3 with aloe vera plant extract (Fig. 1). It was found that the reaction time and temperature had significant effects on the formation of Ag nanostructures. AgNPs were observed as spherical particles with the sizes between 70.7–192.02 nm, moreover the sizes of the materials were significantly affected by their preparation temperature as presented in Table 1. At 6 h, the AgNPs showed the sizes of 70.70 ± 23, 79.47 ± 22, and 161.66 ± 53 nm prepared at 100 °C, 150 °C and 200 °C, respectively. At 12 h, the AgNPs showed sizes of 95.25 ± 23, 149.55 ± 47 and 192.02 ± 53 nm prepared at 100 °C, 150 °C and 200 °C, respectively. Furthermore, TEM image of AgNPs prepared at 100 °C for 6 h indicating that the size of AgNPs was in good agreement with SEM results, UV-vis absorption spectra of AgNPs showed that the maximum absorption was found at 420 nm and was attributed to the surface plasmon resonance of AgNPs (Figs. 2A and 2B) The XRD patterns of AgNPs resulted from using the above 3 hydrothermal conditions (Figs. 3A and 3B). All of the main peaks were indexed as AgNPs with the face centered cubic (fcc) lattice of silver, as shown in the standard data (JCPDS file No.01-071-4613). The diffraction peaks at 2θ degree of 38.2, 44.3, 64.5 and 77.1 corresponded to the (111), (200), (220), and (311) planes, respectively. A pure phase of Ag was only obtained at a temperature of 200 °C for 6 h. The chemical reaction to form a pure phase at 100 and 150 °C for 6 h was incomplete because reaction at such a low temperature usually requires a longer time (Fig. 3A). The existence of Ag2O was shown at the peak at around 31.9. (Liu et al., 2010). The result showed a pure Ag phase in all the samples prepared using hydrothermal conditions for 12 h (Fig. 3B).

Figure 1 SEM images of silver nanoparticles on a glass slide after incubation at different temperature and time combinations.

SEM images of AgNPs were obtained at (A) 100 °C for 6 h, (B) 150 °C for 6 h, (C) 200 °C for 6 h, (D) 100 °C for 12 h, (E) 150 °C for 12 h and (F) 200 °C for 12 h.

Table 1 Sizes of AgNPs and antibacterial efficiency of AgNPs in different hydrothermal processes.

		Inhibition zone diameter (cm)	
AgNPs samples	Size of AgNPs (nm)	S. epidermidis (gram positive bacteria)	P. aeroginosa (gram negative bacteria)	
100 °C—6 h	70.70 ± 22	3.65 ± 0.50*	3.90 ± 0.42*	
150 °C—6 h	79.47 ± 22	1.70 ± 0.43	1.60 ± 0.28	
200 °C—6 h	161.66 ± 53	1.50 ± 0.42	1.40 ± 0.32	
100 °C—12 h	95.25 ± 23	1.72 ± 0.42	1.44 ± 0.29	
150 °C—12 h	149.55 ± 47	3.60 ± 0.56*	3.15 ± 0.49*	
200 °C—12 h	192.02 ± 53	3.90 ± 0.84*	3.45 ± 0.21*	
Notes.

* p < 0.01 compared with an AgNO3 control.

Figure 2 TEM image and UV-vis absorption spectra of AgNPs synthesized using an aloe vera plant-extract solution.

TEM image of AgNPs was obtained at 100 °C for 6 h (A) and UV-vis absorption spectra of AgNPs were shown in the maximum absorption at 42 nm (B).

Antibacterial effects

An advantage of silver nanoparticles is that they are known to have an antibacterial effect (Rai et al., 2012). However, the AgNPs formed during the aloe vera hydrothermal method, AgNPs@AV, need to have bioactive functions. It is especially important to understand the functional effects on microorganisms in order to develop novel antibacterial agents. To demonstrate this activity, AgNPs were studied for their bactericidal effect against pathogenic gram-positive S. epidermidis and gram-negative P. aeruginosa. These two strains are the opportunistic bacteria causing of nosocomial infection, moreover there are the virulent factors involving with antibiotic resistance (Otto, 2009; Livermore, 2002). Thus, our new product might be a material of choice to apply in antimicrobial activity instead of antibiotics. This was done using a qualitative antibacterial well diffusion assay and studying AgNPs interaction with bacteria using SEM. Quantitative antibacterial concentrations were evaluated by determining the minimum bactericidal concentration (MBC). It was observed that the inhibition zones of both pathogens were significant for 0.1 mg/mL AgNPs prepared at 100 °C for 6 h, 150 °C and 200 ° C for 12 h compared with the control (Fig. 4 and Table 1). The AgNPs synthesized under different conditions provided varying bactericidal effects. Then, the effects of two AgNPs@AV samples including those prepared at 100 °C for 6 h and 200 °C for 12 h, were selected for further studies using SEM and MBC. The interaction of AgNPs and microorganisms was shown using SEM. The result indicated the cell membrane changed when contacted with the nanoparticles (Fig. 5). This was particularly true for gram negative bacteria, showing a thin layer of membrane and having pores. Subsequently, MBC of the bacterial concentration at 108 CFU/ml was determined for both S. epidermidis and P. aeruginosa. This demonstrated the lowest concentration of nanoparticles with bactericidal effect was 0.01 mg/mL for AgNPs fabricated at 100 °C for 6 h and 200 °C for 12 h against 108 CFU/ml S. epidermidis. The corresponding concentrations was 0.0025 mg/mL for AgNPs fabricated at 100 °C for 6 h and 0.00125 mg/mL for those formed at 200 °C for 12 h against 108 CFU/ml P. aeruginosa. Moreover, the microbicidal activity of nanoparticles provided high efficiency within 2 months. At the lower AgNPs concentrations, clearly there was an effect on the lethality against gram negative bacteria whereas higher concentrations were needed to control gram positive bacteria.

Figure 3 XRD patterns of AgNPs synthesized using an aloe vera plant-extract solution.

The AgNPs were prepared at temperatures of 100, 150, and 200 °C and for different times (A) 6 h and (B) 12 h. .

Figure 4 Antibacterial activity assay of AgNPs against S. epidermidis and P. aeruginosa.

(A) AgNO3 and aloe-vera extract control in S. epidermidis, (B) AgNO3 and aloe-vera extract control in P. aeruginosa, (C) 100 °C—6 h, 150 °C—6 h, and 200 °C—6 h AgNPs at (0.1 mg/mL) in S. epidermidis, (D) 100 °C—6 h, 150 °C—6 h, and 200 °C—6 h AgNPs at (0.1 mg/mL) in P. aeruginosa, (E) 100 °C—12 h, 150 °C—12 h, and 200 °C—12 h AgNPs at (0.1 mg/mL) in S. epidermidis, (F) 100 °C—12 h, 150 °C—12 h, and 200 °C—12 h AgNPs at (0.1 mg/mL) in P. aeruginosa.

Figure 5 SEM images of the bacterial strains.

(A) S. epidermidis, (B) P. aeruginosa, (C) S. epidermidis treated with 100-6 h AgNPs (0.04 mg/mL), (D) P. aeruginosa treated with 100–6 h AgNPs (0.04 mg/mL).

Figure 6 Illustration of proposed bacterial inactivation mechanism that may involve nanocrystalline AgNPs@AV to disrupt the bacterial membrane.

In the hydrothermal method, various organic compounds such as saponin, tannin, terpenoids, and flavonoids in the aloe vera plant extract can be combined with AgNO3 synthesizing AgNPs@AV. These nanocrystals may accumulate at the cell membrane increasing its permeability, which eventually results in the death of P. aeruginosa and S. epidermidis.

Cytotoxicity evaluation

To determine the cytotoxicity of AgNPs@AV on human cells, PBMCs were tested using the MTT assay. The result was calculated as %survival of the cells cultured with samples at concentrations of 0.04, 0.02, 0.01, 0.005, and 0.0025 mg/mL of 100 °C for 6 h and 200 °C for 12 h processed AgNPs@AV. The %survival of the cells in less 0.0025 mg/mL of both nanoparticles was significantly higher than 50% which confirms that these AgNPs@AVs were non-toxic to human PBMCs. Nanoparticles produced by green synthesis can be useful in biomedical applications.

Discussion

Recently, there has been increasing study of AgNPs synthesis to develop several applications such as catalysis, biosensing, imaging, and antibacterial activity. Green synthesis is an alternative method developed to produce metal nanoparticles by using natural compounds or plant components. These are environmentally friendly processes that avoid the toxicity of chemicals. Algae, bacteria, fungi and plants have been used to synthesize NPs without the need for additional reducing and stabilizing agents. Plant extracts contain functional substances, including cyclic peptides, sorbic acid, citric acid, euphol, polyhydroxy limonoids, ascorbic acid, retinoic acid, tannins, ellagic acid, and gallic acid, among others, are strongly believed to play a crucial role in the bioreduction and stabilization of nanoparticles (Rajan et al., 2015). These processes seem facile, safe, low cost, and ecofriendly, eliminating the elaborate process of maintaining aseptic cell cultures and are suitable for large scale production. Therefore, this study focused on the biosynthesis of AgNPs with plant extracts of aloe vera leaves. Zhang et al. (2010) speculated that the hydroquinones in the aloe vera plant extract act as the reducing agents. Additionally, the spherical shape of AgNPs was governed by the weaker binding of proteins in the solution leading to the isotropic growth of the AgNPs. Here the hydrothermal process was applied to AgNPs synthesis in which time and temperature had an effect on the resulting crystalline structure of AgNPs. High temperature and pressure are necessary to facilitate the reduction processes (Liu et al., 2012). Nucleation and the growth of AgNPs depend on the reaction temperature. Additionally, capping agents also play a role in the synthesis of nanoparticles. Selective interaction of capping agents may lead to anisotropic crystalline growth. Poly (vinyl) pyrrolidoneis are widely used to synthesize nanorods due to their preferential interaction with the (100) plane (Pal, Tak & Song, 2007). In the case of aloe vera, a (111) plane of AgNPs predominantly arose as a major peak. This plane was reported responsible for a strong antibacterial effect (Feng et al., 2000)

The factors controlling the morphology, size, and product purity in the hydrothermal process were reaction temperature and time (Byrappa & Adschiri, 2007; Liu et al., 2014). Moreover, biosynthesis of inorganic nanoparticles with the plant extracts improved their bactericidal effect (Yousefzadi, Rahimi & Ghafori, 2014). High bactericidal activity was possibly caused by synergistic antibacterial effects of AgNPs and naturally-occurring chemicals in aloe vera. The lethal mechanism against pathogenic S. epidermidis and P. aeruginosa might involve the release of Ag+ ions from AgNPs and the formation of crystalline bio-organic compounds of aloe vera plant extract assembled with AgNPs anchored onto the bacterial cell walls, producing pits and penetrating into the cytoplasm. Various natural ligands can interact with microbial membrane such as saponin, tannin, terpenoids, and flavonoids in the aloe vera (Griffin et al., 1999; Sahu et al., 2013). The interaction with the cell membrane may increase its permeability leading to cell lysis. Moreover the free radicals from metal result in induction of oxidative stresses, such as reactive oxygen species (ROS), that can damage the bacterial membranes, mitochondria, and DNA. This eventually results in the death of the cell (Hajipour et al., 2012; Tamboli & Lee, 2014). From our results, a schematic mechanism involving the reaction of AgNPs@AV to kill the bacteria was purposed and illustrated in Fig. 6. Additionally, the susceptibility of different types of bacteria was attributed to the structure of their bacterial cell walls. Previous studies indicated that the silver ion released from AgNPs was responsible for antibacterial activity (Feng et al., 2000). The free silver ion can then bind with the thiol groups of enzymes (Zhang et al., 2013). The AgNPs formed at 100 °C for 6 h were found to be toxic to both grampositive and gramnegative bacteria. This might due to the smaller size of the AgNPs fabricated under these conditions which results a higher surface area (Cui et al., 2013). Silver ion release is a size dependent process (Cui et al., 2013). The antibacterial activity of the synthesized AgNPs might be due to the silver ion release and the resulting genotoxic activity of aloe vera on E. coli (Zhang et al., 2010). Interestingly, the samples processed at 200 °C for 12 h had the largest size of those examined and they provided effective growth inhibition of the pathogens. The results indicated that the larger AgNPs might contain high levels of incorporated aloe vera compounds as well as a pure Ag phase due their long time and high temperature treatment. Therefore, this hybrid nanostructure formed under specific conditions can potential be an antibacterial agent.

Conclusion

This report described a green and facile method to synthesize AgNPs in large quantities. Silver nitrate was reduced in an aloe vera plant-extract solution under a hydrothermal condition. Aloe vera plant extract solutions were used as both reducing and stabilizing agents. Fine spherically shaped nanoparticles were obtained. The particle size of AgNPs can be tuned by varying the hydrothermal temperature. The antibacterial effect of AgNPs@AV showed promise for use as a highly potent agent with minimal cytotoxicity to human PBMCs. These hybrid nanomaterials could potentially be used in biomedical applications.

Supplemental Information

Supplemental Information 1 UV-Vis absorbance spectra

Click here for additional data file.

Supplemental Information 2 XRD analysis of AgNPs-aloe vera synthesized by hydrothermal method

Click here for additional data file.

The authors acknowledge the National Research University Project of Thailand, Office of the Higher Education Commission, through the Advanced Functional Materials Cluster of Khon Kaen University.

Additional Information and Declarations

Competing Interests

Author Contributions

Data Availability

The authors declare there are no competing interests.

Patcharaporn Tippayawat and Apiwat Chompoosor conceived and designed the experiments, performed the experiments, analyzed the data, contributed reagents/materials/analysis tools, wrote the paper, prepared figures and/or tables, reviewed drafts of the paper.

Nutthakritta Phromviyo and Parichart Boueroy conceived and designed the experiments, performed the experiments, analyzed the data, wrote the paper, prepared figures and/or tables, reviewed drafts of the paper.

The following information was supplied regarding data availability:

The raw data has been supplied as Supplementary File.

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
