# Peer review of "Green synthesis of silver nanoparticles in aloe vera plant extract prepared by a hydrothermal method and their synergistic antibacterial activity"

_PeerJ, doi:10.7717/peerj.2589_

## Round 0.1 · original submission · Minor Revisions

· Academic Editor

Minor Revisions

Dear Patcharaporn,

Expert reviewers of the field have reviewed your paper and they have raised some criticisms on your work.

I invite you to carefully respond all the questions raised and resubmit the revised paper.

Best regards,
Maria Rosaria

Reviewer 1 ·

Basic reporting

The manuscript deals with a trendy topic, because of the worldwide interest in silver nanoparticles (AgNPs) and in natural bioactive substances. Unfortunately the AgNPs production process exhibits a broad range of toxicity in vertebrates and invertebrates, and to avoid the chemical toxicity, biosynthesis (green synthesis) of metal nanoparticles is proposed as a cost-effective and environmental friendly alternative. Aloe vera leaf extract is an interesting natural medicinal agent with multiple properties, including an antibacterial effect.

Experimental design

The used methods are correct and effective and the obtained results demonstrate an
antibacterial activity on S. epidermidis and P. aeruginosa. The results showed that AgNPs had a high antibacterial effect, which depended on their synthesis conditions, particularly when processed at 100 oC for 6 h and 200 oC for 12 h. The cytotoxicity of AgNPs was determined using human PBMCs revealing no obvious cytotoxicity.

Validity of the findings

the work provides interesting and innovative findings, with applicative potential.

Comments for the author

In general I think that a larger number of microbial species should has to be tested, so I think the Authors have to explain why they tested only two microbial strains and why they tested those two strains. The Authors have to consider and underline this aspect when they discuss the obtained results.

Reviewer 2 ·

Basic reporting

No Comments

Experimental design

No Comments

Validity of the findings

No Comments

Comments for the author

Patcharaporn Tippayawat et al. report Green synthesis of silver nanoparticles in aloe veraplant extract prepared by a hydrothermal method and their synergistic antibacterial activity. The results are impressive especially for the antibacterial activity section. The article is suitable for this journal according to the content aspect. I recommend this article for publication.
Nevertheless, I have some suggestions which are listed below.
1. It is better to provide some representative TEM images of the obtained Ag nanoparticles because the AgNP in the SEM images in this study were aggregated from which the diameter measurement seems difficult.
2. If it is possible please provide the UV absorption spetra of the obtain colloid because the surface plasmon resonance band (PRB) of the Ag colloid can indicate some structural information.

·

Basic reporting

In the following some criticisms and weakeness are summarized.

Some works from literature have been carried out focused on green biosynthesis of silver nanoclusters with antimicrobial functionalities, some of them are also cited into the manuscript. Authors must indicate with clarity what the gap of knowledge their results have been covering beyond he state of the art.

Dose-relationships between the concentration of the silver nanoclusters (as obtained at 100°C and 6h as well as at 200°C and 12h) and load of vital microbial cell are strongly recommended to be reported and discussed in depth in the manuscript.

For a more easy reading, the manuscript should report the state of the art in the introduction section only.

Experimental design

Authors should report more information about the experimental method used to estimate MIC levels as well as on how they accounted for the medium turbidity due to the presence of the silver nanoclusters during the growth/inhibition tests.

Validity of the findings

A storage stability study about the antimicrobial ability for the two types of the more effective silver nanoclusters is lacking. No data are provided about nano structures and residual antimicrobial ability during storage intended for biotechnological or medical applications

---

## Round 0.2 · Minor Revisions

· Academic Editor

Minor Revisions

Dear Authors,

Reviewer 3 has commented that your response to them that: "In the experiment, MIC is the lowest concentration of drug that inhibits the growth of the organism that it was observed the turbid or clear solution after the incubation time and also compared with broth alone control. Our results could provide the clear solution in the dose of MIC due to the killing bacteria aggregated with the nanoparticles underneath the broth solution. Furthermore, the sampling supernatants of each condition were measured with spectrophotometer" is not clear enough.

As you can see from their review, they would like you to respond more fully to this concern.

Reviewer 1 ·

Basic reporting

The manuscript deals with a trendy topic, because of the worldwide interest in silver nanoparticles (AgNPs) and in natural bioactive substances. Unfortunately the AgNPs production process exhibits a broad range of toxicity in vertebrates and invertebrates, and to avoid the chemical toxicity, biosynthesis (green synthesis) of metal nanoparticles is proposed as a cost-effective and environmental friendly alternative. Aloe vera leaf extract is an interesting natural medicinal agent with multiple properties, including an antibacterial effect.

Experimental design

The used methods are correct and effective and the obtained results demonstrate an
antibacterial activity on S. epidermidis and P. aeruginosa. The results showed that AgNPs had a high antibacterial effect, which depended on their synthesis conditions, particularly when processed at 100 oC for 6 h and 200 oC for 12 h. The cytotoxicity of AgNPs was determined using human PBMCs revealing no obvious cytotoxicity.

Validity of the findings

The work provides interesting and innovative findings, with applicative potential.

Comments for the author

The Authors have considered the given suggestions, mentioning at least the matter concerning the choice of testing only a limited number of microbial species.

·

Basic reporting

Authors provided point-by-point responses to each of the comments with some minor weakeness.

Experimental design

No comments

Validity of the findings

Please make clear again the way you provide evidence of the inhibitory (reversible) and microbicidal (irreversible) effects of nanoclusters against the two investigated strains
A recovery procedure and subsequent incubation test might provide evidence of reversibility/irreversibility of nanocluster aggregation observed at 0.01mg/mL and 0.0025mg/mL for S. epidermidis and P. aeruginosa strains. Otherwise, you will provide evidence by microscopy of cell membrane damage (pores) for the S. epidermidis at 0.01 mg/mL or for that of the P. aeruginosa one at 0.0025mg/mL.

Comments for the author

Please make the following minor changes.

At line 109 - please change “vera” with “vera”
At line 202 - please change “stain” with “strains”

---

## Author Rebuttal · Round 0.2

Division of Clinical Microbiology,

Faculty of Associated Medical Sciences,

Khon Kaen University, Thailand 40002

Department of Chemistry,

Faculty of Science, Ramkhamhaeng University,

Bangkok, Thailand 10240

                                                                July 1st, 2016

Dear Editors

We thank the reviewers for their generous comments on the manuscript "Green synthesis of silver nanoparticles in aloe vera plant extract prepared by a hydrothermal method and their synergistic antibacterial activity"

Those comments of the reviewers were highly insightful and enabled us to greatly improve the quality of our manuscript. In the following pages are our point-by-point responses to each of the comments of the reviewers as well as your own comments.

Please find attached a revised version of our manuscript, which we believe that the manuscript is now suitable for publication in PeerJ.

We shall look forward to hearing from you at your earliest convenience.

Yours sincerely,

Dr. Patcharaporn Tippayawat

Lecturer of Clinical Microbiology

Dr. Apiwat Chompoosor

Assistant Professor of Chemistry

On behalf of all authors

*Reviewer 1 (Anonymous)*

*Basic reporting*

*The manuscript deals with a trendy topic, because of the worldwide interest in silver nanoparticles (AgNPs) and in natural bioactive substances. Unfortunately the AgNPs production process exhibits a broad range of toxicity in vertebrates and invertebrates, and to avoid the chemical toxicity, biosynthesis (green synthesis) of metal nanoparticles is proposed as a cost effective and environmental friendly alternative. Aloe vera leaf extract is an interesting natural medicinal agent with multiple properties, including an antibacterial effect.*

*Experimental design*

*The used methods are correct and effective and the obtained results demonstrate an antibacterial activity on S. epidermidis and P. aeruginosa. The results showed that AgNPs had a high antibacterial effect, which depended on their synthesis conditions, particularly when processed at 100 ºC for 6 h and 200 ºC for 12 h. The cytotoxicity of AgNPs was determined using human PBMCs revealing no obvious cytotoxicity.*

*Validity of the findings*

*The work provides interesting and innovative findings, with applicative potential.*

*Comments for the Author*

*In general I think that a larger number of microbial species should has to be tested, so I think the Authors have to explain why they tested only two microbial strains and why they tested those two strains. The Authors have to consider and underline this aspect when they discuss the obtained results.*

Agree with the point that you are concerned. Therefore, we have added in discussion with the point of the importance of the two stains.

In page 9 line 214-217, "These two stains are the opportunistic bacteria causing of nosocomial infection, moreover there are the virulent factors involving with antibiotic resistance *(Otto 2009, Livermore 2002)*. Thus, our new product might be a material of choice to apply in antimicrobial activity instead of antibiotics"

*Reviewer 2 (Anonymous)*
*Basic reporting*
*No Comments*

*Experimental design*
*No Comments*

*Validity of the findings*
*No Comments*

***Comments for the Author***

*Patcharaporn Tippayawat et al. report Green synthesis of silver nanoparticles in aloe veraplant extract prepared by a hydrothermal method and their synergistic antibacterial activity. The results are impressive especially for the antibacterial activity section. The article is suitable for this journal according to the content aspect. I recommend this article for publication. Nevertheless, I have some suggestions which are listed below.*

*1. It is better to provide some representative TEM images of the obtained Ag nanoparticles because the AgNP in the SEM images in this study were aggregated from which the diameter measurement seems difficult.*

*2. If it is possible please provide the UV absorption spetra of the obtain colloid because the surface plasmon resonance band (PRB) of the Ag colloid can indicate some structural information.*

Thank you very much for this suggestion. Therefore we have already added these information in content and figure 2 in page 19.

1. The TEM image was added to the revised manuscript in page 8 line 191-193. The result was consistent with SEM images.

2. The absorption spectra were added to the revised manuscript in page 8 line 193-195 indicating that the maximum absorption was found at 420 nm and was attributed to the surface plasmon resonance of AgNPs.

*Reviewer 3 (Pasquale Massimiliano Falcone)*

*Basic reporting*

*In the following some criticisms and weakness are summarized.*

*Some works from literature have been carried out focused on green biosynthesis of silver nanoclusters with antimicrobial functionalities, some of them are also cited into the manuscript. Authors must indicate with clarity what the gap of knowledge their results have been covering beyond the state of the art.*

*Dose-relationships between the concentration of the silver nanoclusters (as obtained at 100°C and 6h as well as at 200°C and 12h) and load of vital microbial cell are strongly recommended to be reported and discussed in depth in the manuscript.*

In this regard, the bactericidal doses of the silver nanoparticlers were presented by MBC at 0.01 mg/mL for AgNPs fabricated at 100ºC for 6h and 200ºC for 12h against $10^8$ CFU/ml *S. epidermidis*. The corresponding concentration was 0.0025 mg/mL for AgNPs fabricated at 100ºC for 6h and 0.00125 mg/mL for those formed at 200ºC for 12h against $10^8$ CFU/ml *P. aeruginosa*. For this point was shown in the result part of antibacterial effects (page 9-10 line 228-237).

Also it was discussed in the part of discussion (page 12 line 294-303) that the AgNPs formed at 100ºC for 6h and 200ºC for 12h were found to be toxic to both gram-positive and gram-negative bacteria. It might due to the smallest size of 100ºC for 6h AgNPs fabricated under these conditions which results a higher surface area to release the silver ion and aloe vera on bacteria. Whereas, the samples processed at 200ºC for 12h had the largest size containing more amount of organic compounds, therefor it provided obviously effective growth inhibition of *P. aeruginosa* providing at lower two times of AgNPs fabricated at 200ºC for 12h when compared with AgNPs fabricated at 100ºC for 6h.

*For a more easy reading, the manuscript should report the state of the art in the introduction section only.*

Following this suggestion, the state of the art was emphasized that "this work has shown the use of naturally occurring compounds to be a reducing and stabilizing agent. This method is considered green synthesis. The resulting silver nanoparticles showed a synergism of aloe verla and silver nanoparticles on bactericidal effect. This hybrid nanomaterial provides an alternative material for using in antibacterials." and addressed in page 4-5 line 101-104.

*Experimental design*

*Authors should report more information about the experimental method used to estimate MIC levels as well as on how they accounted for the medium turbidity due to the presence of the silver nanoclusters during the growth/inhibition tests.*

Thank you very much to give us the notice in this point. In the experiment, MIC is the lowest concentration of drug that inhibits the growth of the organism that it was observed the turbid or clear solution after the incubation time and also compared with broth alone control. Our results could provide the clear solution in the dose of MIC due to the killing bacteria aggregated with the nanoparticles underneath the broth solution. Furthermore, the sampling supernatants of each condition were measured with spectrophotometer.

*Validity of the findings*

*A storage stability study about the antimicrobial ability for the two types of the more effective silver nanoclusters is lacking. No data are provided about nano structures and residual antimicrobial ability during storage intended for biotechnological or medical applications.*

Agree with the important point. We have tested the reproducible activity of the nanoparticles by using well diffusion method that it was measured on diameter of the inhibition zone (mm) revealing the same inhibition zone for two months after the nanoparticles synthesis which it has already addressed in page 10 line 237-238.

---

## Round 0.3 · accepted · Accept

· Academic Editor

Accept

Dear Prof. Patcharapon,

Congratulations!
Your paper is now ready to be published in PeerJ.

Best regards

---

## Author Rebuttal · Round 0.3

Division of Clinical Microbiology,

Faculty of Associated Medical Sciences,

Khon Kaen University, Thailand 40002

Department of Chemistry,

Faculty of Science, Ramkhamhaeng University,

Bangkok, Thailand 10240

September 17$^{th}$, 2016

Dear Editors

We thank the reviewers for their generous comments on the manuscript "Green synthesis of silver nanoparticles in aloe vera plant extract prepared by a hydrothermal method and their synergistic antibacterial activity"

Those comments of the reviewers were highly insightful and enabled us to greatly improve the quality of our manuscript. The following pages are our point-by-point responses to each of the comments of the reviewers as well as your own comments in the second revision.

Please find attached a revised version of our manuscript, which we believe that the manuscript is now suitable for publication in PeerJ.

We shall look forward to hearing from you at your earliest convenience.

Yours sincerely,

Dr. Patcharaporn Tippayawat

Lecturer of Clinical Microbiology

Dr. Apiwat Chompoosor

Assistant Professor of Chemistry

On behalf of all authors

*Reviewer 1 (Anonymous)*

*Basic reporting*

*The manuscript deals with a trendy topic, because of the worldwide interest in silver nanoparticles (AgNPs) and in natural bioactive substances. Unfortunately the AgNPs production process exhibits a broad range of toxicity in vertebrates and invertebrates, and to avoid the chemical toxicity, biosynthesis (green synthesis) of metal nanoparticles is proposed as a cost-effective and environmental friendly alternative. Aloe vera leaf extract is an interesting natural medicinal agent with multiple properties, including an antibacterial effect.*

*Experimental design*

*The used methods are correct and effective and the obtained results demonstrate an antibacterial activity on S. epidermidis and P. aeruginosa. The results showed that AgNPs had a high antibacterial effect, which depended on their synthesis conditions, particularly when processed at 100 oC for 6 h and 200 oC for 12 h. The cytotoxicity of AgNPs was determined using human PBMCs revealing no obvious cytotoxicity.*

*Validity of the findings*

*The work provides interesting and innovative findings, with applicative potential.*

*Comments for the author*

*The Authors have considered the given suggestions, mentioning at least the matter concerning the choice of testing only a limited number of microbial species.*

Thank you very much for these suggestions.

***Reviewer (Pasquale Massimiliano Falcone)***

***Basic reporting***

*Authors provided point-by-point responses to each of the comments with some minor weakeness.*

***Experimental design***

*No comments*

***Validity of the findings***

*Please make clear again the way you provide evidence of the inhibitory (reversible) and microbicidal (irreversible) effects of nanoclusters against the two investigated strains. A recovery procedure and subsequent incubation test might provide evidence of reversibility/irreversibility of nanocluster aggregation observed at 0.01mg/mL and 0.0025mg/mL for S. epidermidis and P. aeruginosa strains. Otherwise, you will provide evidence by microscopy of cell membrane damage (pores) for the S. epidermidis at 0.01 mg/mL or for that of the P. aeruginosa one at 0.0025mg/mL.*

Thank you very much for your suggestion. It is very important issue. We therefore perform the antibacterial activity of two high effective nanoparticles including 100°C for 6h and 200°C for 12h of AgNPs@AV at the concentrations of 0.01 mg/mL and 0.0025 mg/mL, respectively. The electron micrographs show the evidence of the particles damaging bacterial membrane as shown in SEM micrographs in the figure below. These results suggest that irreversible effects of the nanoclusters in the each different concentrations against the two pathogens.

**Untreated microbial control**
***P. aeruginosa***

**AgNPs@AV 100ºC for 6h**

**0.04 mg/mL**    **0.0025 mg/mL**

[Figure] [Figure]

**AgNPs@AV 200ºC for 12h**

**0.04 mg/mL**    **0.0025 mg/mL**

[Figure] [Figure]

***S. epidermidis***

**AgNPs@AV 100ºC for 6h**

**0.04 mg/mL**    **0.01 mg/mL**

[Figure] [Figure] [Figure]

**AgNPs@AV 200ºC for 12h**

**0.04 mg/mL**    **0.01 mg/mL**

[Figure] [Figure]

***Comments for the author***

*Please make the following minor changes.*

*At line 109 - please change "vera" with "vera"*

*At line 202 - please change "stain" with "strains"*

Thank you very much for the prove reading. The words have been already corrected.

Changing the word "verla" with "vera" at line 99.

Changing the word "stains" with "strains" at line 202.